# Two oncomiRs, miR-182-5p and miR-103a-3p, Involved in Intravenous Leiomyomatosis

**DOI:** 10.3390/genes14030712

**Published:** 2023-03-14

**Authors:** Edyta Barnaś, Joanna Ewa Skręt-Magierło, Sylwia Paszek, Ewa Kaznowska, Natalia Potocka, Andrzej Skręt, Agata Sakowicz, Izabela Zawlik

**Affiliations:** 1Institute of Health Sciences, Medical College of Rzeszow University, Kopisto 2a, 35-959 Rzeszow, Poland; 2Institute of Medical Sciences, Medical College of Rzeszow University, Kopisto 2a, 35-959 Rzeszow, Polandizazawlik@gmail.com (I.Z.); 3Laboratory of Molecular Biology, Centre for Innovative Research in Medical and Natural Sciences, College of Medical Sciences, University of Rzeszow, 35-959 Rzeszow, Poland; 4Department of Medical Biotechnology, Medical University of Lodz, Żeligowskiego 7/9, 90-752 Lodz, Poland

**Keywords:** intravenous leiomyomatosis, epigenetic factors, oncomiRs, miR-182-5p, miR-103a-3p

## Abstract

Leiomyomas, also referred to as fibroids, belong to the most common type of benign tumors developing in the myometrium of the uterus. Intravenous leiomyomatosis (IVL) tends to be regarded as a rare type of uterine leiomyoma. IVL tumors are characterized by muscle cell masses developing within the uterine and extrauterine venous system. The underlying mechanism responsible for the proliferation of these lesions is still unknown. The aim of the study was to investigate the expression of the two epigenetic factors, oncomiRs miR-182-5p and miR-103a-3p, in intravenous leiomyomatosis. This study was divided into two stages: initially, miR-182-5p and miR-103a-3p expression was assessed in samples coming from intravenous leiomyomatosis localized in myometrium (group I, *n* = 6), intravenous leiomyomatosis beyond the uterus (group II; *n* = 5), and the control group, i.e., intramural leiomyomas (group III; *n* = 9). The expression level of miR-182-5p was significantly higher in samples coming from intravenous leiomyomatosis (group I and group II) as compared to the control group (*p* = 0.029 and *p* = 0.024, respectively). In the second part of the study, the expression levels of the studied oncomiRs were compared between seven samples delivered from one woman during a four-year observation. The long-term follow-up of one patient demonstrated significantly elevated levels of both studied oncomiRs in intravenous leiomyomatosis in comparison to intramural leiomyoma samples.

## 1. Introduction

The term uterine smooth muscle tumors (USMTs) refers to a group of histological, genetic, and clinical heterogeneous tumors located in the uterus.

USMT classification does not include tumors originating from myometrium but of extrauterine location. This group of tumors is known as leiomyoma beyond the uterus (LBU) and includes disseminated peritoneal leiomyomatosis (DPL), benign metastazing leiomyoma (BML), intravenous leiomyomatosis (IVL), and parasitic leiomyoma (PL) located in the broad ligament or retroperitoneal space [1,2]. The mechanism of DPL origin, which spreads benign leiomyomatous tumors on the surface of the peritoneal cavity, distinguishes them from the abovementioned group [2]. According to Ondulu et al., DPL in its course develops through metaplasia of mesenchymal cells [3].

Intravenous leiomyomatosis is considered a rare type of uterine leiomyoma. Although the condition has histologically benign features, it is clinically aggressive, being characterized by the intraluminal growth of leiomyomas in intrauterine and systemic veins. These lesions are implants from coexisting or previously resected uterine fibroids. Few studies are available of this subject, and most of them include case reports, with approximately 170 cases of IVL described [1,2]. Intravenous spread is another hypothetic dissemination mechanism, which may also be the case in the rest of the groups. According to the classification of Ma et al., some groups of LBU are the stages of intravenous spread. Intravascular growth named intravenous leiomyomatosis (IVL) starts in small veins of the uterus body and then advances to the veins in the parametria. The next step of the intravenous leiomyomatosis proliferation is the invasion of renal veins, right atrium and ventricle, pulmonary artery, and lungs [4].

The process of metastasis of benign leiomyoma demonstrates to some extent a similarity to malignant tumors consisting of intravasation, proliferation, and extravasation. Some epigenetic factors are involved in this process in malignant tumors.

Several studies on various tumors have investigated miRs and their relationship with metastasis. Among the microRNAs with oncogenic activity is miR-182. It affects proliferation and promotes cancer metastasis and epithelial–mesenchymal transition (EMT). It is interesting to note that miR-182 may be both an oncogene and a cancer suppressor. Abnormal levels of miR-182 expression have been linked to several cancers, for example, ovarian cancer [5], breast cancer [6], hepatocellular cancer, [7], prostate cancer [8], hepatocellular carcinoma [9], and others. Most of these studies selected the miRs analyzed based on previous studies, often involving other cancers. However, Sachdeva’s study was based on the wide spectrum of miRs with increased expression of miR-182 found in sarcoma, a malignant mesodermal tumor. This project was accomplished in vivo on a mouse model, and demonstrated that a single miRNA can regulate the metastasis of primary tumors by coordinated regulation of multiple genes [10].

Another microRNA involved in the ability to induce the epithelial–mesenchymal transition of cells is miRNA-103 (miR-103), a member of the miR-103/107 family. MiR-103 is implicated in some biological and pathological processes, including being associated with several cancers. Upregulated levels were found in endometrial cancer tissues compared to neighboring non-cancerous tissues [11]. Similarly, upregulated expression was noted in cervical cancer [12], hepatocellular carcinoma [13], and colorectal cancer [14]. The selection of miRs in our study was based on data from the literature, therefore, we selected miR-103, which is involved in the metastasis of breast cancer. The second investigated miR was 182-5p, which is involved in the metastasis of many cancers [15,16,17,18]. Both oncomiRs levels are elevated in cells expressing estrogen receptors and both undergo the regulation by estrogen [19,20,21]. Moreover, both miRNAs are involved in carcinogenesis. Although the role of miR-182 in the process of uterine leiomyoma development was documented elsewhere [22], its implication in the formation of extrauterine fibroids is still unknown. As in the case of miR-182, there are no reports available on miR-103 and the development of leiomyomatosis beyond the uterus. Considering the fact that miR-103 enhances the process of intravasation and metastasis of various cancers, such as colorectal, hepatocellular, and prostate, and its implication in the development of the female reproductive tract, it would be useful to analyze the status of miR-103 in the leiomyomatosis of an unusual localization [23,24,25,26].

This retrospective, longitudinal study investigates the association between miRNA (miR-182-5p and miR-103a-3p) expression levels and the successive spreading of leiomyomatosis beyond the uterus during four subsequent surgeries. These procedures were described in in our previous study [2].

The research hypothesis was that both miRNAs could be implicated in the process of migration and proliferation of leiomyoma into the distant tissues localized beyond the uterus. Moreover, we intended to check whether the process of intravasation into uterine vessels might be linked with the dysregulation of expression of miR-182 and miR-103.

The aim of the study was to assess miR-182-5p and miR-103a-3p in the deparaffinized samples derived from intravenous leiomyomatosis beyond the uterus, intravenous leiomyomatosis located in myometrium, and intramural leiomyoma.

## 2. Materials and Methods

### 2.1. Samples

Twenty samples taken from twelve patients were analyzed in the study. They included eleven samples of intravenous myoma obtained from six patients and nine intramural leiomyoma samples received from seven women. Among them, one patient delivered seven samples (three intravenous and four intramural leiomyomatosis samples) during the four-year observation.

Material from one patient was obtained over 4 years. During the first operation, two fibroids were excised from the endometrium. The following year, the myoma was removed from the parametria. Over the next years, myomas from the uterus, retroperitoneal space, and parametria were removed, and in the last year a myoma located in the inferior vena cava was excised. An extensive description is included in the report by Barnaś et al. [2].

Moreover, another woman delivered two samples: one intramural and one intravenous leiomyomatosis.

All fibroid samples were divided taking into account two criteria:Localization: intra the uterus or beyond the uterus;Localization: intravenous or beyond the vein.

All fibroid samples were divided according to their localization into three study groups:

Group I—intravenous leiomyomatosis located in the myometrium;

Group II—intravenous leiomyomatosis located beyond the uterus;

Group III—intramural leiomyomas located in the myometrium, i.e., the control group.

Tissue sections used for the study were routinely fixed in 10% buffered formalin, pH 7.2 to 7.4, for 24–48 h, then automated using the Thermo Scientific Excelsior Tissue Processor and paraffin embedded. Microscopic slides were cut at 3 microns and stained with hematoxylin and eosin in a Thermo Scientific Varistain Gemini Automated Slide stainer. In each case, smooth muscle actin, estrogen receptor, progesterone receptor, and desmin and h-caldesmon immunohistochemical stains were performed using Roche Ventana Benchmark Ultra machines. Each preparation was subject to histopathological evaluation. In the microscopic image, tumor cellularity, cell shape, nuclear atypia, dividing activity, stroma hyalinization, and the presence of necrosis were assessed. In all cases, the tissue was composed of bundles of spindle-shaped cells without nuclear atypia, dividing, and necrotic figures, with intravascular growth. The stroma showed varying degrees of glazing. Immunohistochemical staining showed a positive reaction for smooth muscle actin, estrogen receptor, progesterone receptor, desmin, and h-caldesmon (Figure 1). No leiomyoma was found in the vicinity.

Human samples were acquired from the Department of Pathology under a protocol approved by the Ethics Committee of the University of Rzeszow, no.11/10/2015. 

The adopted study protocol was designed in two parts. The first part concerned the comparison of miR-182-5p and miR-103a-3p expression levels between the three subpopulations of leiomyomatosis (groups I, II, and III). In the second part of the study the miR-182-5p and miR-103a-3p expression levels were compared between intramural leiomyomas samples (*n* = 3) and intravenous leiomyomatosis samples (*n* = 4) obtained from one patient.

### 2.2. Total RNA Isolation and RT-qPCR

Total RNA including miRNA was isolated from FFPE tissue (formalin-fixed, paraffin-embedded) using miRNeasy FFPE Kit (Qiagen, Valencia, CA, USA) in accordance with the manufacturer’s protocol. The first step was to deparaffinize the samples using xylene. Then, the samples were treated with lysis buffer with proteinase K. DNase was used to remove the DNA. The RBC buffer and ethanol were added to the resulting lysates to ensure better RNA binding of the column. The next step was to purify the sample using an RNeasy MinElute spin column. Elution of total RNA, including miRNA, was evaluated in 14 mL of RNase-free water. RNA concentration and purity were quantitated using a NanoDrop Spectrophotometer (NanoDrop Technologies, Wilmington, DE, USA). cDNA was synthesized from 10 ng of total RNA using the TaqMan™ MicroRNA Reverse Transcription Kit (Applied Biosystems, Foster City, CA, USA) according to the protocol, which was then used for quantitative real-time polymerase chain reaction (qRT-PCR). miRNA quantification was performed using TaqMan™ Universal PCR Master Mix (Applied Biosystems) and TaqMan™ MicroRNA Assay (Applied Biosystems): hsa-miR-182-5p (Assay ID 002334), hsa-miR-103a-3p (Assay ID 000439), and RNU6B (Assay ID 001093). RNU6B was used as an endogenous control. The 20 μL PCR included 5 μL template cDNA, 10 μL TaqMan Universal PCR Master Mix, and 1 μL TaqMan miRNA Assay (20×). These reactions were performed in the following conditions: 50 °C for 2 min, 95 °C for 10 min, followed by 55 cycles of 95 °C for 15 s and 60 °C for 1 min. Real-time PCR analysis was performed on a Cobas Z480 (Roche, Basel, Switzerland) with LightCycler 480 Software (Roche).

### 2.3. Statistical Analyses

Statistical analyses were performed using Statistica 13.1 (StatSoft). The normal distribution of continuous data was verified with the Shapiro–Wilk test. Normally distributed data were further analyzed with Student’s *t*-test, while multivariable analyses of not normally distributed data were analyzed with the Kruskal–Wallis and post hoc Dunn’s tests. The correlations between the age of the patients and relative expression levels were calculated with the Spearman’s test. The Wilcoxon signed-rank test for the matched pairs was performed. The correlation between the age and the expression levels was analyzed with Spearman rank correlation test. A *p*-value below 0.05 for the test results was considered statistically significant.

## 3. Results

### 3.1. Analysis of the Level of Expression between the Study Groups

In the first part, we compared the analyzed miRNA expression in the intravenous leiomyomatosis localized in the myometrium (group I), intravenous leiomyomatosis localized beyond the uterus (group II), and intramural leiomyomas (group III).

The total number of collected samples and the age of women at the time of surgery are presented in Table 1. The mean age of the women suffering from the intramural leiomyomas was insignificantly lower with respect to those with leiomyomatosis beyond the uterus (38.9 vs. 43.6; *p* > 0.05; Student’s *t*-test).

The age of the women significantly correlated with the relative miR-182-5p expression (R_Spearman_ = 0.646; *p* = 0.002; Figure 2A), whereas no correlation between the expression of miR-103a-3p and the age of patients was observed (R_Spearman_ = 0.163; *p* = 0.491; Figure 2B).

The difference in miR-182-5p expression levels for all analyzed samples was observed between intramural leiomyomas vs. intravenous leiomyomatosis located in the myometrium and between intramural leiomyomas vs. intravenous leiomyomatosis beyond the uterus (Figure 3A). These analyses suggest that miR-182-5p is engaged both in the process of leiomyomatosis invasion into the uterus vessels as well as the process of intravenous leiomyomatosis proliferation and expansion. These differences were not observed with respect to miR-103a-3p (Figure 3B).

The association between the miR-103a-3p expression and atypical localization of leiomyomatosis was not observed in the present study. This finding might suggest that miR-103a-3p is not involved in both the processes of fibroid intravasation and leiomyomatosis proliferation and expansion.

### 3.2. Analysis of the Level of Expression within a Single Case

In the second part of our study, the analyses of miRNA expression between samples coming from one case were performed. Seven leiomyomatosis samples of various localizations were obtained from one patient from a four-year follow-up period. The locations of these samples were as follows: intramural leiomyomas—three samples removed during the first surgery, and intravenous leiomyomatosis—four samples (one intravenous leiomyomatosis sample located in the myometrium and three samples included of intravenous leiomyomatosis beyond the uterus), removed during the next surgeries. The Wilcoxon signed-rank test pointed that both miR-182-5p and miR-103a-3p were significantly lower in the intramural leiomyomas as compared to intravenous leiomyomatosis (Figure 4A,B). This might suggest that both studied miRNAs are involved in the leiomyomatosis proliferation process.

The study of one patient revealed that the differences in miR-182-5p and miR-103a-3p expression levels were not observed between intramural leiomyomas and intravenous leiomyomatosis in the uterus (for both analyses *p* = 0.109) and between intravenous leiomyomatosis located in the myometrium and intravenous leiomyomatosis beyond the uterus (*p* = 0.108 and *p* = 0.592 for miR-182-5p and miR-103a-3p, respectively).

## 4. Discussion

MicroRNAs are single-stranded RNA molecules with a length of 21 to 23 nucleotides. Their role is to regulate gene expression at the post-transcriptional level. MiRNAs influence the expression of several genes and are therefore crucial for critical cellular processes such as proliferation, cell cycle, differentiation, apoptosis, or cancer transformation [27]. A modified miRNA expression profile has been demonstrated in various diseases. A single molecule can regulate up to 200 different mRNA target molecules. This indicates the importance of miRNAs in the regulation of many cellular processes, both physiological and pathological [28].

Moreover, miRNAs are highly tissue specific and play significant roles in cell proliferation and differentiation. As a result, dysregulated miRNA expression can result in cell dedifferentiation, oncogenesis, tumor metastasis. and tumor invasion [29].

The post-transcriptional regulation of gene expression by miRNA and its association with the carcinogenesis was discussed in numerous studies [27,28,29,30]. MiRNA, as short fragments of non-coding RNA, are implicated in the degradation of mRNA, influencing the depletion in the level of the protein product coded by the degraded mRNA. Some studies indicate that miRNAs are also implicated in uterine leiomyoma development [31,32,33]. However, the literature provides only limited information about the influence of miRNA on the processes of benign leiomyoma spreading. The report by Nuovo et al. suggested that benign metastasizing leiomyomas are indeed most likely benign lesions, and upregulation of miR-221 expression is an accurate way to differentiate leiomyosarcoma from benign metastasizing leiomyoma [34].

By adopting the thesis that intravenous spread is responsible for benign leiomyoma metastasis, the present study explores the association of miRNA (miR-182-5p and miR-103a-3p) with the development of LBU. The clinical and immunohistochemical observation of our patients suffering from LBU allows us to claim that intravenous spread of leiomyomatosis would be divided into two stages. The first step is related to the process of intravasation. The leiomyoma starts to secrete some factors, e.g., metalloproteinases into the local environment, influencing the process of degradation of extracellular matrix [35]. Consequently, the leiomyoma might arise in the vascular wall, where fibroid tissue is covered by endothelium. In the case of malignant tumors, intravasation is followed by “cross-talk” between tumor cells and the venous or lymphatic endothelium [36]. Many factors are involved in this interaction, including miRNAs [37,38]. The fact that metastatic myoma is covered by endothelium makes it impossible to be treated as a foreign body inside the vessel and to avoid thrombus formation. Moreover, it facilitates the nutrition and gas exchange of a myoma in the veins and the heart, which can take place through the peduncle or through the endothelium, as it happens, e.g., in the placenta.

Several studies proved that miR-182 might be a potential biomarker of cancer development. Its elevated expression is observed both in cancer tissues (e.g., breast cancer, prostate cancer, or glioblastoma) as well as in cancer cell lines (e.g., MB-231 or MCF-7 cell lines) [39,40]. Interestingly, results of numerous studies indicated that this oncomiR might play a dual role in the development of cancers, and this role depends on the cancerous tissue. The overexpression of genes coding for miR-182-5p was observed in melanoma, colon cancer, prostate cancer, and in cells of endometrial cancer. This oncomiR downregulates the expressions of tumor suppressor genes, i.e., FOXO3, MITF, FOXO1, or MTSS1, and thus promotes the process of cancerogenesis [9,41,42]. Moreover, miR-182-5p regulates the process of angiogenesis by the inhibition of KLF2 and KLF4, i.e., genes whose products suppress the activity of receptor 2 for vascular endothelial growth factors or the maintenance of the integrity of the endothelial barrier. This proangiogenic effect was observed both for patients suffering from glioblastoma and in vitro studies [43]. Additionally, miR-182-5p mediates the downregulation of genes implicated in DNA repair, thus effects on the cell cycle, apoptosis, and genetic stability supporting tumorgenesis [42]. Similar results were observed for bladder cancer, where miR-182-5p represses the expression of Cofilin 1 and the loss of this miR activity promotes tumor cell proliferation, migration, and invasion as well as tumorigenesis abilities regulated by Cofilin 1 [44]. However, some studies recognized miR-182-5p as an inhibitor of the process of proliferation; indeed, miR-182-5p downregulates the mRNA products for proteins promoting the cancer development, including CTTN, RGS17, and CREB1 in ovarian papillary carcinoma, gastric adenocarcinoma, and in lung cancer [5,45,46].

The human and animal studies reveal that the natural estrogens have a significant impact on the elevated level of miR-182 in tumors of the reproductive track [10,47]. Fibroids as benign tumors develop in the uterus and become the main diagnostic challenge in differentiation between them and uterine sarcoma. Both types of tumors are involved in uterine mesodermal tumors, and it is well documented that miR-182 plays a significant role in the development of uterine sarcoma, which was observed in the sarcoma samples of human tissue and in a mouse sarcoma model [48]. The study of Sachdeva et al. demonstrated that the deletion in miR-182 is related to the inhibition of te intravasation process of sarcoma cells into circulation [10].

As sarcoma and fibroids are classified as mesenchymal tumors, the question arose whether miR-182 could be also related to the intravasation and metastasis of leiomyoma. To the best of our knowledge, the present study is the first one to convey the information that both intravenous leiomyomatosis beyond the uterus and those located in the uterus present a higher expression level of miR-182 compared to intramural leiomyomas. This might suggest that, as in case of sarcoma tissue, the process of intravasation of fibroids is regulated by miRNAs, including miR-182, as it was observed by Sachdeva et al. that this oncomiR is responsible for the post-translational regulation of genes coding metalloproteinases. Additionally, in this study based on ELISA and gelatin zymography techniques, they observed that both MMP-2 and MMP-9 have reduced activity in the media from miR-182-deleted sarcoma cells [10].

Moreover, significant elevation of miR-182-5p in comparison to intravenous fibroids was observed in our study, which allows us to believe that this oncomiR could also be engaged in the process of leiomyoma metastasis. Consistent with this claim, Yu et al. observed that miR-182 targets SMAD7 to potentiate TGF-β-induced metastasis [49]. The association between TGF-β and the development of leiomyoma was discussed numerous times in the literature, and this molecule or its receptors seem to be significant targets for a design of new therapies [50,51]. 

The next oncomiR whose level was examined in the present study was miR-103a-3p. It was observed in numerous studies that this molecule is associated with cancer development and progression [15,16,17,18,52,53]. In hepatocellular carcinoma cell lines, it was found that miR-103 promotes epithelial–mesenchymal transition. However, there is no proof that this oncomiR is linked to the process of intravasation [54]. In the present study, we did not observe any relationship between the process of intravasation of fibroids and the level of miR-103a-3p. Additionally, when we compared the miR-103a-3p level between intravenous leiomyomatosis located in the uterus or beyond the uterus and intramyometrial tissue, we also did not find any association. This might suggest that both miRNAs are not related to the progression and development of fibroids. A completely new perspective was provided by the analysis comparing the miR-103a-3p expression level between the fibroids collected from different localizations but which were delivered by one case. This analysis has shown that the level of miR-103a-3p is significantly higher in the fibroid tissue localized beyond the uterus than that localized in a typical location. In the first part of our investigation, we obtained all fibroid samples from perimenopausal women. On the contrary, the samples collected for the second part of the study belonged to a single patient, aged 34.

This might also explain the contrary observations regarding miR-103: on the one hand it is responsible for the inhibition of cancer proliferation and migration, whereas others link this oncomiR with the development of tumors.

The mechanism of action of miR-103 in different cancers and various cases seems ambiguous. In colorectal cancer, miR-103 promotes cancer stemness [55]. In leukemias, it inhibits both proliferation and apoptosis during GC treatment [56].

It is of value that this study includes the reference to the case of a patient operated on several times due to LBU myomas migrating in various tissues. This allows for longitudinal evaluation of the selected microRNAs.

It would also be worth analyzing the blood sample in the future in order to find a biomarker of this condition.

### Limitations

This study has some limitations. First, a low number of patients qualified for this study, being a consequence of the rarity of incidences of intravenous leiomyomatosis. This limits or just prevents conducting a prospective, randomized study. Second, the study was based on the analysis of only two miRNAs, not full miRNA signatures; indeed, in the pathomechanism of intravenous leiomyomatosis, a lot of miRNAs are implicated. Additionally, in the present study, only the association between the intravenous leiomyomas and the two oncomiRs miR-182-5p and miR-103a-3p was investigated. The influence of these oncomiRs on the pathomechanism of the disease was not developed. We believe that these oncomiRs support the process of development of leiomyomas outside of uterine tissue by the regulation of genes implicated in this process. However, we did not study the gene expression levels being potentially targeted by these oncomiRs, which is the third limitation of this study. Moreover, an in vitro model of the action of miR-182-5p or miR-103a-3p on the regulation of molecular pathways in the leiomyoma cells was not examined. Therefore, the results of the present study should be considered as an introduction to the exploration of the knowledge about the rare process of the invasion of leiomyoma outside of uterine tissue, and a further study based on in vitro and animal models should be considered in the future.

## 5. Conclusions

In conclusion, this study was the first to demonstrate the relation of miR-182-5p and miR-103a-3p with the intravenous spread of benign leiomyoma. MiR-182-5p could be related to the processes of intravasation and intravenous proliferation. Moreover, we observed that the role of miR-103a-3p in the process of intravenous spread of leiomyomatosis might be associated with the age of patients and therefore their hormonal status.

## Figures and Tables

**Figure 1 genes-14-00712-f001:**
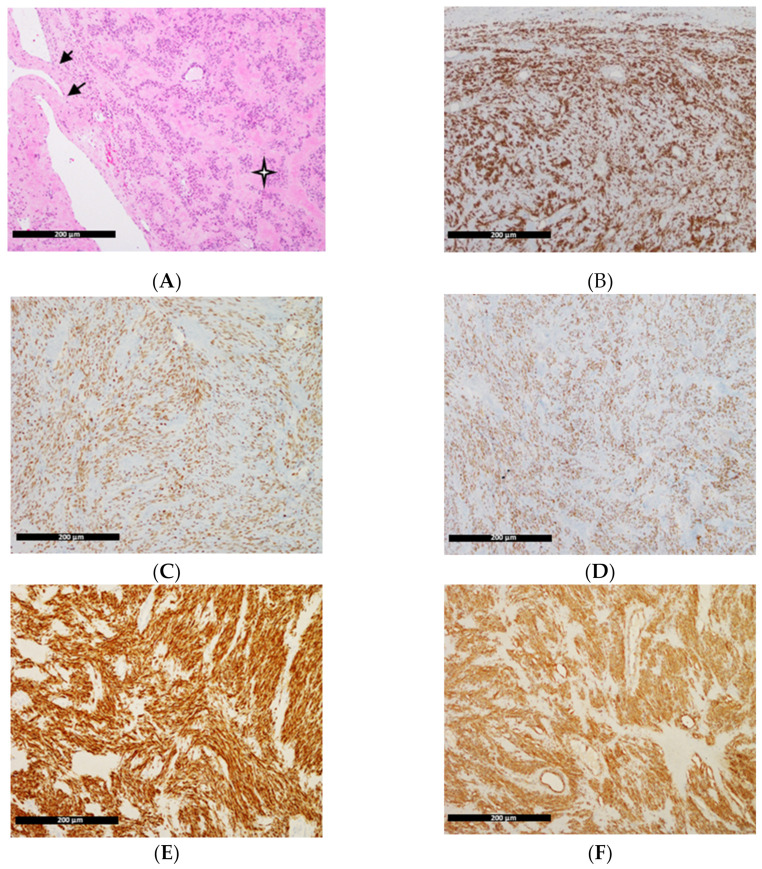
Microscopic pictures of intravenous leiomyomatosis. (**A**) The mass of tumor 
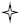
 with a peduncle 
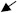
. Objective 10×, H&E. The scale bar showed 200 μm in distance. Immunohistochemical features of intravenous leiomyomatosis beyond the uterus. (**B**) Strong positivity for smooth muscle actin. (**C**) Estrogen receptors. (**D**) Progesterone receptors. (**E**) Desmin. (**F**) H-caldesmon. Objective 10×.

**Figure 2 genes-14-00712-f002:**
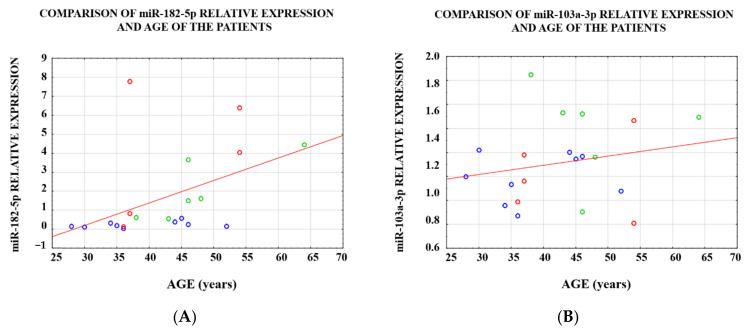
Correlation of miR-182-5p expression (**A**) and miR-103a-3p expression (**B**) with the age in 3 groups: group I (*n* = 6, green circle), group II (*n* = 5, red circle), and group III (*n* = 9, blue circle).

**Figure 3 genes-14-00712-f003:**
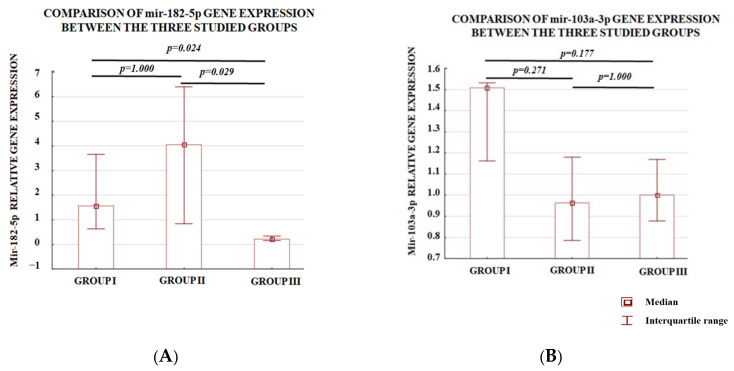
The expression of miR-182-5p (**A**) and miR- 103a-3p (**B**) in three groups: group I—intravenous leiomyomatosis located in the myometrium, group II—intravenous leiomyomatosis beyond the uterus, and group III—intramural leiomyomas. The *p*-values were calculated by the use of Kruskal–Wallis with post hoc Dunn’s test.

**Figure 4 genes-14-00712-f004:**
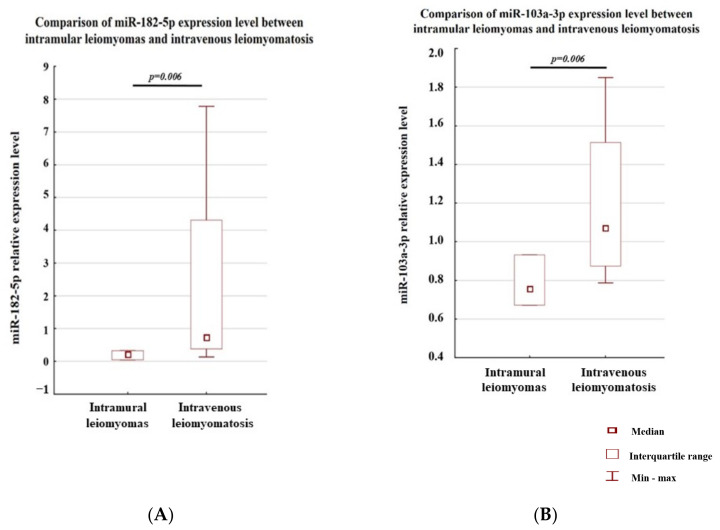
Expression of miR-182-5p (**A**) and miR-103a-3p (**B**) in intramural leiomyomas samples (*n* = 3) and intravenous leiomyomatosis (*n* = 4) from one patient.

**Table 1 genes-14-00712-t001:** The number of samples and the age of the studied women (*n* = 20).

Localization	Number of Cases	Mean Age (Years) ± Standard Deviation
intravenous leiomyomatosis located in myometrium—group I	6	47.5 ± 9
intravenous leiomyomatosis beyond the uterus—group II	5	43.6 ± 9
intramural leiomyomas—group III	9	38.9 ± 8

## Data Availability

The data presented in this study are available on request from the corresponding authors.

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
