# Peer review of "Two oncomiRs, miR-182-5p and miR-103a-3p, Involved in Intravenous Leiomyomatosis"

_genes, 2023, doi:10.3390/genes14030712_

Round 1
Reviewer 1 Report
-Although this study is interesting, there were still some question to concern.
It is an interesting study, since the patients number is limited in this manuscript, the level of miR-182-5p and miR-103a-3p in blood should be observed since the intravenous leiomyomatosis is coming form venous.
- If there were significant increase, it could be a useful blood biomarker of leiomyoma.
1.This manuscript had investigated the expression of the two epigenetic factors: oncomiRs: miR-182-5p 22 and miR-103a-3p and got interesting finding, but the previous other reports illustrate there were a lot of microRNA were related with the proliferation and the process of metastasis, the reason and more evidence to select this two oncomiRs should be extensively discussed.
2.the tissue sample were used to measured and isolate total RNA, whether there are process to avoid the contamination of blood cell and other tissues.
3.both oncomiRs were increase in intravenous leiomyomatosis, no matter it is located in myometrium or beyond the uterus, it might illustrate that there was no organic specificity. The intravenous leiomyomatosis in other organ's should be observed or discussed based on the previous reports, such as” Diagn Mol Pathol. 2008 Sep; 17(3): 145–150”
4.for differentiating the intravenous leiomyomatosis from intramural leiomyomas, the blood sample is easier to use in clinical work, if it is possible, the blood results of this two oncomRs should be observed.
Author Response
Dear Reviewer, First of all, we would like to thank you very much for your interest, and time spent reviewing our manuscript. We appreciate the valuable comments which helped us to create a much more appropriate work:
Reviewer:
-Although this study is interesting, there were still some question to concern. It is an interesting study, since the patients number is limited in this manuscript, the level of miR-182-5p and miR-103a-3p in blood should be observed since the intravenous leiomyomatosis is coming form venous. If there were significant increase, it could be a useful blood biomarker of leiomyoma.
The subject of our study was to determine mirs in tissues. This idea of marking them in the blood is interesting and may be an indication for further research, to potentially be a biomarker of leiomyoma, as in the paper from 2022 on colorectal cancer.: Raut JR, Bhardwaj M, Niedermaier T, Miah K, Schrotz-King P, Brenner H. Assessment of a Serum Microrna Risk Score for Colorectal Cancer among Participants of Screening Colonoscopy at Various Stages of Colorectal Carcinogenesis. Cells. 2022 Aug 8;11(15):2462. doi: 10.3390/cells11152462. PMID: 35954306; PMCID: PMC9367813.
1.This manuscript had investigated the expression of the two epigenetic factors: oncomiRs: miR-182-5p 22 and miR-103a-3p and got interesting finding, but the previous other reports illustrate there were a lot of microRNA were related with the proliferation and the process of metastasis, the reason and more evidence to select this two oncomiRs should be extensively discussed.
We agree that a lot of microRNAs were related to the proliferation and the process of metastasis, and like the authors of other works, we have dealt with selected microRNAs. In addition, we have included a broader description of Sachdev's study in the main text of the paper, where it was shown in an animal model that mir-182 deletion reduces circulating tumour cells. When over-expression of mir-128 increases circulating tumour cells. In Sachdeva report there are important conclusion based on in vivo studies that a single microRNA can regulate metastasis. We have expanded the explanation in the paper that it is an animal model and that to some extent the data of the animal model can be transferred.
In addition, we have added an explanation that this applies to other cancers, i.e. colorectal cancer, hepatocellular, prostate.
2.the tissue sample were used to measured and isolate total RNA, whether there are process to avoid the contamination of blood cell and other tissues
Thank you for your suggestions and comments. To avoid contamination of other tissue, the tissue for RNA isolation was subjected to histopathological evaluation. We have included a detailed explanation in the text of the manuscript:
“Tissue sections used for the study were routinely fixed in 10% buffered formalin, pH 7.2 to 7.4, for 24-48 hours, then automated using the Thermo Scientific Excelsior Tissue Processor and paraffin embedded. Microscopic slides were cut at 3 microns and stained with hematoxylin and eosin in a Thermo Scientific Varistain Gemini Automated Slide stainer. In each case, smooth muscle actin, estrogen receptor, progesterone receptor, desmin and h-caldesmon immunohistochemical stains were performed using Roche Ventana Benchmark Ultra machines. Each preparation was subject to histopathological evaluation. In the microscopic
image, tumor cellularity, cell shape, nuclear atypia, dividing activity, stroma hyalinization and the presence of necrosis were assessed. In all cases, the tissue was composed of bundles of spindle-shaped cells without nuclear atypia, dividing and necrotic figures, with intravascular growth. The stroma showed varying degrees of glazing. Immunohistochemical staining showed a positive reaction for smooth muscle actin, estrogen receptor, progesterone receptor, desmin, and h-caldesmon. No leiomyoma was found in the vicinity.”
3.both oncomiRs were increase in intravenous leiomyomatosis, no matter it is located in myometrium or beyond the uterus, it might illustrate that there was no organic specificity. The intravenous leiomyomatosis in other organ's should be observed or discussed based on the previous reports, such as” Diagn Mol Pathol. 2008 Sep; 17(3): 145–150”
LBUs are rare and it is impossible to evaluate them in different organ locations. However, it would be very interesting to see if location changes expression. In our previous study, we addressed the localization of BML [18]. It is interesting that Nuovo also reports mirRNA 221 up-regulation and that their expression is accurate way to differentiate leiomyomasarcoma from BML. We have also included this in the text of our work.
4.for differentiating the intravenous leiomyomatosis from intramural leiomyomas, the blood sample is easier to use in clinical work, if it is possible, the blood results of this two oncomRs should be observed.
We agree that blood would be a very useful material if the existence of BML in a particular patient had been known prior to surgery. It may be a potential issue for further research, as we mentioned at the end of the discussion.

Reviewer 2 Report
In each case we performed a histological examination (H&E, SMA, estrogen receptor, 97 progesterone receptor, desmin, h-caldesmon) as in our previous studies [2,18]. 98
2.Barnaś, E.; Raś, R.; Skręt-Magierło, J.; Wesecki, M.; Filipowska, J.; Książek, M.; Skręt, A.; Widenka K. Natural history of 312 leiomyomas beyond the uterus. Medicine (Baltimore). 2019,98,e15877. 

18.Raś, R.; Książek, M.; Barnaś, E.; Skręt- Magierło, J.; Kądziołka, W.; Fudali, L.; Filipowska, J.; Skręt, A. Benign metastasizing 345 leiomyoma in triple location: lungs, parametria and appendix. Menop. Rev. 2016,15,1-5.
Comments: You have mentioned in the above paragraph as in our previous studies. When you write method in the paper you must briefly mention the method here also.

The examples of intravenous leiomyomatosis were composed of solid sheets of well-differentiated smooth muscle cells. The stroma revealed hyaline changes. 99
Comment: What do you mean by examples , do you mean criteria??
It differs from its malignant 100 counterpart by lack of atypia and mitotic activity (Figure 1). 101
Total RNA including miRNA was isolated from FFPE tissue using miRNeasy FFPE 118
Comment: Kindly elaborate FFPE
leiomyomatosis beyond the uterus), removed during the next surgeries. The Wilcoxon 185
Comment: You have mentioned next surgeries, kindly elaborate how many surgeries. This will also help us to understand recurrence.
A numerous studies indicate- 201
Comment: You can write Various studies instead of A
5.Sachdeva, M.; Mito, J.K.; Lee, C.L.; Zhang, M.; Li, Z.; Dodd, R.D.; Cason. D.; Luo, L.; Ma,Y.; Van Mater, D.; Gladdy, R.; Lev, D.C.; Cardona, D.M.; Kirsch, D.G. MicroRNA-182 drives metastasis of primary sarcomas by targeting multiple genes [pub-
lished correction appears in J. Clin. Invest. 2016,126,1606]. J. Clin. Invest. 2014, 124,4305-4319.
Comment: You cannot quote two studies as one reference like reference 5.
This might also explain the contrary observations regarding mir-103; once it is re- 270 sponsible for inhibition of cancer proliferation and migration whereas others link this on- 271 comir with the development of tumors [36-38]. 272
Comment: I did not understand this paragraph, could you kindly clarify.
The subjects of our study were epigenetic factors in IVL. At the end of our report it 273 would be interesting to present the review of current literature dealing with genomic al- 274 ternations in IVL. The authors: Buza et al.[39], Ordulu et al.[40], Zhang et al.[41], Ordulu 275 et al.[42] performed theirs studies on concise material of IVL. The subject of their studies 276 were: oligonucleotide array comparative genomic hybridization [39], expression of 277 HMGA2, MDM2,and CDK4 proteins by immunohistochemistry [40], RNA- sequencing 278 on IVL tumors [41], array comparative genomic hybridization (aCGH) and Cyclin D1, 279 p16,phosphorylated-Rb, SMARCB1, SOX10,CAIX, SDHB and FH immunohistochemistry 280 [42]. The results of above studies were not conclusive. They found only differences in frequencies of studied parameters. 281
Comment: Do you really think reference 39-42 is required for your manuscript
Line :273 to281
Reviewer 3 Report
1. It would be better for the readers’ to understand the scientific question if the authors can add more information regarding intravenous leiomyomatosis, such as the incidence frequency for introduction part.
2. What are the inclusive/exclusive criteria of the recruited patients? More details should be addressed clearly in the method section.
3. Scale bar should be added in Figure 1.
4. The primer sequence for RT-qPCR should be listed in the M&M, which could make the results repeatable.
5. The author should explore the expression level of miR-182-5p and miR-103a-3p in at least two dependent cases, which will lead a convincing conclusion without any individual bias.
6. This manuscript only focused on the association relationship between the expression level of the two oncomiRs, more works (such as knockdown/inhibition or overexpression) should be explored to establish the causal relationship in a scientific rigid way. Based on these, the author should give a deeply discussion about the limitations.
Reviewer 4 Report
The authors conducted a retrospective longitudinal study to investigate the association between two miRNAs 70 (miR-182-5p and miR-103a-3p) expression levels and the development of leiomyomatosis beyond the uterus.
The research provides novel insight into the process of intravenous leiomyomatosis and is of high interest to the gynecological community. The basis for the research was studies that investigated the expression of the mentioned miRNAs in the carcinogenesis of other malignancies and mesenchymal tumors. The main limitation is the number of subjects as a result of the rare incidence of intravenous leiomyomatosis. Nevertheless, the paper has a high scientific quality and may serve as a basis for larger, maybe even multicenter studies, to further investigate the involvement of two miRNAs' in this process.
However, several points require further revision.
Introduction; lines 52 and 53: Add a reference for this claim.
Discussion; line 233: "As sarcoma and fibroids are classified as mesenchymal malignancies". How can fibroids be classified as malignancy?
Discussion; lines 239 and 240: Reference required, even though the following sentence has one.
Discussion; lines 253 and 254: Reference required.
Discussion; lines 260 to 269. First of all, the sentence "Interestingly, as the case of the woman analyzed in this paper is special as fibroids were diagnosed in all our patients at the perimenopausal period, the first fibroids extraction from this woman took place when she was 34" needs to be modified. I suggest the following: "In the first part of our investigation, we obtained all fibroid samples from perimenopausal women. On the contrary, the samples collected for the second part of the study belonged to a single patient, aged 34". Moreover, I think that the claim that miR-103a-3p might be involved in the spread of fibroids in younger women, at this point, is just speculation based on the expression in a single patient. Therefore, I advise You to remove this part of the discussion.
Discussion; lines 273 to 282: I advise providing more details from the mentioned studies. It is excellent that You wanted to include the short literature review, but You should highlight the results from the studies with more detail.
Round 2
Reviewer 3 Report
For the last concerned (the 6th) that I raised before, the author should add the corresponding statement in the discussion part (Limitations) for the revised manuscript. Besides that, other the points that I raised have been addressed properly.
